# Amphiphilic Janus Microspheres Prepared by Caged Photoactivatable Alkoxysilane

**Wei Li** [1,2,*], **Daodao Hu** [2], **Jinxin Ma** [3] and **Shukun Shen** [2]

1. Shaanxi Engineering Technology Research Center of Controllable Neutron Source, School of Science, Xijing University, Xi'an 710123, China
2. School of Materials Science and Engineering, Shaanxi Normal University, Xi'an 710100, China; daodaohu@snnu.edu.cn (D.H.); shukun_shen@snnu.edu.cn (S.S.)
3. National Energy Ningxia Coal Industry Co., Ltd., Coal Chemical Industry Technology Research Institute, Yinchuan 750004, China; 15054215@chnenergy.cn
* Correspondence: 20200002@xijing.edu.cn

**Abstract:** A simple photolysis route was proposed to prepare Amphiphilic Janus Particles (AJP) based on SiO$_2$ microspheres. The surface of SiO$_2$ microspheres were modified by photoactive alkoxysilane, which was synthesized by dealcoholization condensation of 6-nitroveratroyloxycarbonyl and isocyanatopropyl-triethoxysilane. UV irradiation caused eater-breaking allowed for the precise control of hydrophilic modification of the hemispherical exposed particles surfaces. The component and morphology of the obtained particles were characterized by fourier transform infrared spectroscopy and ultraviolet-visible spectroscopy, and the Janus feature was evaluated by scanning electron microscopy, transmission electron microscopy, and dispersity in the oil–water dual-phases. The following results were obtained. The AJP with 450 nm size processes the hydrophilic amino groups on one side and the hydrophobic 6-nitroveratryloxycarbonyl moieties on the other. Additionally, the AJP were located at the phase boundary between water and n-hexane, and the negative charged gold nanoparticles with 25 nm size were adsorbed only onto the side with the positive charged amino groups. The AJP have interfacial adsorption energies that can be as much as three times larger than that of homogeneous particles and thus exhibit excellent surface activities.

**Keywords:** amphiphilic Janus particles; UV-irradiation; photoactive alkoxysilane; golden nanoparticles; asymmetrical distribution

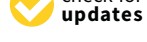



## 1. Introduction

Janus particles, first coined by Pierre-Gilles de Gennes in his 1991 Nobel Lecture, are named after the two-faced Roman god to depict the particles with two different properties on each side [1]. Since the concept was presented, the various Janus particles with different surface components, structures or compartments on both sides have attracted extensive interest for their anisotropic chemical, optical, electronic, and magnetic properties [2–4]. Even this concept has been extended to the field of dendrimers with asymmetric structures. Recently, the multifunctional amphiphilic Janus dendrimers have been extraordinarily successful for the targeted delivery of mRNA to provide COVID-19 vaccines [5,6]. Besides, a method to detect the anti-SARS-CoV-2 spike antibody level by using Janus emulsions or Janus particles as biosensors was reported [7].

Owing to the special structure of Janus particles, which always exhibits unique characters. The geometry of particles, especially those deviating from perfectly spherical shapes, is of significant importance in colloidal assembly. This is because the geometry influences the particle "recognition", determines the particle packing, and ultimately dictates the formation of assembled superstructures [8]. The amphiphilic Janus particles (AJP) of this article, composed of hydrophilic and hydrophobic hemispheres, are some of the simplest

anisotropic colloids, and they exhibit higher surface activities than particles with homogeneous surface properties. AJP have interfacial adsorption energies that can be as much as three times larger than that of homogeneous particles and thus exhibit excellent surface activities [9,10].

To obtain Janus particles with a desired function, a large variety of synthetic methods have been developed to contribute to the thriving growth of Janus particle research. Considering applications, AJP with particular chemistries are more important among these Janus particles because the chemical composition plays a key role in determining the functions. Generally, AJP can be prepared by selective modification of one side of the particles' surfaces [11]. By this way, the particles with one side of reactivity are firstly prepared, and then they are functionalized through the post-reaction with appropriate agents. So, it is significant to prepare AJP with potentially functional groups. To fabricate AJP with the available functional groups, many strategies have been reported, such as temporary masking [12], microcontact printing [13], partial contacting with active medium [14], based grafting polymerization at an interface [15], and the use of directional flows or fields [16].

In view of the local orientation and induced modification, photo-irradiation possesses superiority in the preparation of AJP. Ultraviolet light (UV) is a well-known directional field and has been extensively applied in photochemistry. The features of UV in directional field and photochemistry make it especially suitable to prepare AJP. Generally, a dissymmetrical surface modification of pristine particles is feasible in virtue of a descent of UV light intensity. It derives mainly from UV absorption by photoactive compounds contained in reactant solutions or anchors to the particles and combines with light sheltering of the lower hemisphere by the upper hemisphere [17]. Q. Wu et al. reported that crescent-moon-shaped microparticles and microcapsules with designated structural characteristics are further produced under ultraviolet light of photopolymerization after removing one hemisphere of the Janus microdroplets [18]. Although UV has been used to introduce polymers on one side of the particle for the preparation of AJP, photo-cleavage reaction involved in this process has been scarcely reported. In fact, photo-cleavable protecting groups (sometimes called photo-releasable, photo-removable or photoactivatable) are very suitably used in the preparation of AJP, because they provide spatial and temporal control over the release of various chemicals. Among the various photo-cleavable protecting groups reported, 6-nitroveratroyloxycarbonyl (NVOC) group, an o-nitrobenzyl derivative, has gained much attention due to its efficient removal upon UV [19,20]. NVOC is a popular nitrobenzyl photolabile protecting group due to its ease in coupling and stability at typical room lighting conditions. NVOC is often used in protecting amino and hydroxyl groups [21]. The reaction scheme is shown as follows (Scheme 1). It can be seen from this scheme that the NVOC protected group was cleaved, leaving a reactive hydroxyl group or amino group.

**Scheme 1.** Reactions of NVOC protected (**A**) hydroxyl group and (**B**) amino group photo-cleaved by UV light.

Inspired by the NVOC protecting strategy, we present a simple method for fabricating functional sub-micron sized $SiO_2$ AJP with chemically reactive amino groups. The corresponding process is presented in Scheme 2. Herein, the photo-active alkoxysilane NVOC-U-PTEOS is utilized as a surface modification reagent on hydrophilic $SiO_2$ particles through the dealcoholization condensation between hydroxyl groups and ethoxyl

groups. Then the SiO$_2$ monodisperse layer is vertically photo-irradiated for a given time. Finally, SiO$_2$ AJP with amphiphilicity could be obtained from one side with hydrophobic 6-nitroveratroyloxycarbonyl groups and the other side with hydrophilic amino groups.

**Scheme 2.** Preparation process flow diagram of Janus particles in this article.

## 2. Materials and Methods

### 2.1. Materials

6-Nitroveratryloxycarbonyl alcohol (NVOC-OH, 98%), isocyanatopropyl-triethoxysilane (NCO-PTEOS, 95%) and di-n-butyltindilanlate (BDTA) were obtained from Alfa. Tetrahydrofuran (THF), n-hexane, trisodium citrate dehydrate, ethanol, NH$_3 \cdot$H$_2$O and methylbenzene were obtained from Aladdin. 3-mercaptopropionic acid (MPA) and HAuCl$_4 \cdot$4H$_2$O were purchased from Aldrich. Both tetraethoxysilane (TEOS) and triethylamine were obtained from Aladdin. TEOS was additionally purified by the fractional distillation at atmospheric pressure, using for the synthesis the fraction, separated at temperature from 167 to 170 °C [22].

### 2.2. Synthesis of NVOC-U-PTEOS

NVOC-U-PTEOS was synthesized according to the literature [23]. The corresponding procedure is shown in Scheme 3.

**Scheme 3.** Reaction process and apparatus for synthesis of NVOC-U-PTEOS.

The typical process for the preparation of NVOC-U-PTEOS as follows was carried out. To 20 mL of THF were added 1.00 g of NVOC-OH, 2.00 mL of NCO-PTEOS, and 0.03 mL of BDTA, and the mixture was refluxed for 24 h under a nitrogen atmosphere. After evaporation of THF, the crude product was washed using hexane three times under nitrogen. The faint yellow precipitate was vacuum drying at 30 °C for 3 h and protected from light.

### 2.3. Preparation of Silica Particles

An improved Werner Stöber's method [24] was used to prepare silica particles. In the process, the $H_2O/CH_3CH_2OH/NH_3$ molar ratio of 3.75/2.4/1.55 was employed. Solution A, containing 100 mL $NH_3 \cdot H_2O$ and 150 mL $CH_3CH_2OH$, was placed in a flask, and then solution B containing 15 mL TEOS and 100 mL $CH_3CH_2OH$ was added dropwise via a constant pressure funnel into solution A at a stirring speed of 200 rpm. The mixture turned turbid after a few minutes, and the mixture was unceasingly stirred for 2 h at room temperature. Finally, a white product was formed. The product was collected by centrifugation (12,000 rpm, 15 min), and was washed by ethanol at least five times. The final precipitate was dried for 4 h at 100 °C under vacuum.

### 2.4. Surface Modification of SiO₂ with NVOC-U-PTEOS

According to Scheme 4, the $SiO_2$ particles with photoactive organosiloxane NVOC-U-PTEOS were obtained by using the reaction between $SiO_2$ particles and NVOC-U-PTEOS. To a 50 mL three-neck flask was added 1.25 g of $SiO_2$ particles, 0.5732 g of NVOC-U-PTEOS and 20 mL of methylbenzene. The mixture was vigorously stirred for 3 min, then 0.36 mL of triethylamine was added as a catalyst.

**Scheme 4.** Reaction between NVOC-U-PTEOS and $SiO_2$.

The mixture was refluxed under a nitrogen atmosphere for 24 h. After that, the mixture was centrifuged at 6000 r/s 15 min, and washed by ethanol for 3 times, then vacuum dried for 3 h at 30 °C. The NVOC-U-PTEOS modified $SiO_2$ particles (named NVOC-U-PTEOS-$SiO_2$) were obtained.

### 2.5. Preparation of AJP by Photo-Cleavage of NVOC-U-PTEOS-SiO₂

According to the scheme for the preparation of AJP, a monolayer NVOC-U-PTEOS-$SiO_2$ was first prepared. Herein, the method for the formation of self-assembled monolayers was employed [25]. The typical process was carried out as follows. The cleaned quartz substrate with dimensions of 55 $cm^2$ was vertically dipped into the suspension of NVOC-U-PTEOS-$SiO_2$ for a few minutes and withdrawn at a programmed speed. The wet coating adhering to the quartz substrate was subsequently air dried under ambient conditions, and the monolayers of NVOC-U-PTEOS-$SiO_2$ on quartz substrate were obtained. In this process, the operational parameters were: the concentration of NVOC-U-PTEOS-$SiO_2$ (10 wt.%), withdrawal speed (3 mm/s), immersion time (4 min) and type of solvent (ethanol/water 1:1 mixture). The temperature was fixed at 20 °C for all the experiments.

As shown in Scheme 2, the prepared monolayers of NVOC-U-PTEOS-$SiO_2$ on quartz substrate was covered by another quartz plate, and one side was vertically photo-irradiated (365 nm, 3.03 $mW/cm^{-2}$) for 10 min [26]. After that, the covered quartz plate was removed, and the microspheres were washed off from the quartz substrate by ultrasonic cleaning in alcohol. The microspheres were collected by centrifugation and were dried in vacuum. The desired AJP, NVOC-U-PTEOS-$SiO_2$-$NH_2$ microspheres particles, were finally obtained. Similarly, $H_2N$-$SiO_2$-$NH_2$ microspheres could be obtained as two sides were vertically photo-irradiated for 10 min.

### 2.6. Characterization

Photo-cleavage: Photo-cleavage reaction of NVOC-U-PTEOS can occur under photo-irradiation. To confirm this reaction, the photo-cleavage reaction of NVOC-U-PTEOS monitored by ultraviolet visible absorption spectrometer (Hitachi Ltd, U-3900/3900H, Tokyo, Japan) was performed. A solution of NVOC-U-PTEOS dissolved in methanol $(5.8 \times 10^{-4}$ M) was added into a quartz cell. The solution was irradiated and was monitored every 20 min by using a UV-vis spectrophotometer. The UV absorption spectra were collected during the 1 h irradiation. In this process, as an ultraviolet source, a CHF-XM35-500 W ultrahigh pressure short arc mercury lamp was used as optical filters with wavelength of 365 nm, and optical power was 3.03 mW/cm$^2$.

Morphology and Contact Angle: Morphologies of SiO$_2$ and NVOC-U-PTEOS-SiO$_2$ were characterized by using FEI scanning electron microscopy (Quanta 200, FEI, Kassel, Germany), using an accelerating voltage of 30 kV. The samples were coated with a thin layer of gold before measurement. Morphologies of SiO$_2$ microspheres, NVOC-U-PTEOS-SiO$_2$ microspheres and NVOC-U-PTEOS-SiO$_2$-NH$_2$ microspheres were also characterized by a high resolution transmission electron microscope (JEM-2100 TEM operating at 200 kV, Jeol Ltd., Tokyo, Japan).

The TEM samples were prepared by placing drops of the corresponding particles suspension on 300 mesh copper grids. In order to observe the distribution of NVOC-U-PTEOS-SiO$_2$ microspheres on the quartz substrate, a copper mesh attached on the quartz substrate was prepared, and the monolayer of NVOC-U-PTEOS-SiO$_2$ microspheres could form on the copper mesh as the same monolayer formed on the quartz substrate. The powders of samples were pressed to prepare tablets, and the contact angles of the tablets were determined using the sessile drop method using a Dataphysics OCA-20 contact angle analyzer (Dataphysics, Filderstadt, Germany).

Janus Feature: To confirm the Janus feature of the asymmetrically photo-irradiated NVOC-U-PTEOS-SiO$_2$, gold nanoparticles were used as a probe to demonstrate NVOC-U-PTEOS-SiO$_2$-NH$_2$ by using the interaction the citrate stabilized AuNPs with negative charge and the NVOC-U-PTEOS-SiO$_2$-NH$_2$ with a positive charge of protonated amino groups on one side [27–30].

An aqueous suspension of citrate-stabilized colloidal gold particles (20–50 nm) was prepared by using the method reported in the literature [31]. The prepared suspension of colloidal gold particles was added to 1 mL suspension of NVOC-U-PTEOS-SiO$_2$-NH$_2$ microspheres under stirring at room temperature for 24 h, and then the mixture was centrifuged (3000 rpm) for 5 min to remove unabsorbed colloidal gold particles. Finally, the desired sample was obtained. For comparison, the same processes for NVOC-U-PTEOS-SiO$_2$ microspheres and the uniformly photo-irradiated NVOC-U-PTEOS-SiO$_2$ microspheres, H$_2$N-SiO$_2$-NH$_2$ microspheres, were carried out. Finally, the morphologies of the prepared samples were characterized by a high resolution transmission electron microscope (JEM-2100 TEM operating at 200 kV, Jeol Ltd., Tokyo, Japan) [32].

Additionally, the Janus feature of NVOC-U-PTEOS-SiO$_2$-NH$_2$ was also verified by using the amphiphilic property. Primary SiO$_2$ microspheres, NVOC-U-PTEOS-SiO$_2$ microspheres and NVOC-U-PTEOS-SiO$_2$-NH$_2$ microspheres were added respectively to a water–hexane dual-phase system, and then were gently shaken. After standing for a few minutes, the dispersion behaviors of the corresponding microspheres in the water–hexane dual-phase system were observed.

## 3. Results and Discussion

### 3.1. Composition of NVOC-U-PTEOS

NVOC-U-PTEOS was synthesized by the reaction between NVOC-OH and NCO-PTEOS. The product was characterized by the FT-IR spectrum and the $^1$H NMR spectrum, and the results are shown in Figure 1.

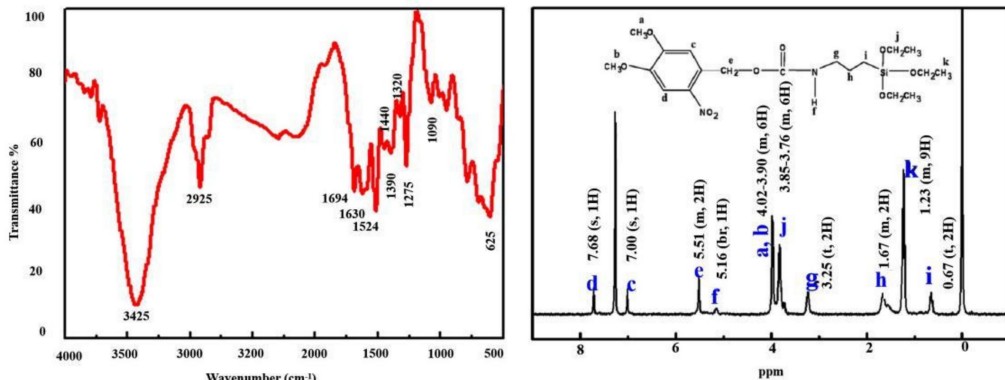

**Figure 1.** FT-IR spectrum (**left**) and [1]H NMR spectrum (**right**) of NVOC-U-PTEOS.

From the IR spectrum, the characteristic peaks [33,34] at 3425 cm$^{-1}$ are attributed to stretching mode of NH, 2926 cm$^{-1}$ associated with CH vibration, 1694 cm$^{-1}$ ascribed to stretching vibration of >C=O, 1630 cm$^{-1}$ related to NHCO, 1524 cm$^{-1}$ and 1320 cm$^{-1}$ designated to NO$_2$ asymmetric and symmetric stretching vibrations, 1440 cm$^{-1}$ and 1390 cm$^{-1}$ ascribed to asymmetric and symmetric deformation modes of CH$_3$, 1275 cm$^{-1}$ assigned to the CH$_2$ wagging mode, 1090 cm$^{-1}$ attributed to Si−O−C stretching mode, 625 cm$^{-1}$ designated to Si-C bond. All these results indicated a successful preparation of NVOC-U-PTEOS. NVOC-U-PTEOS was also characterized by [1]H NMR. The chemical shifts δ (ppm) are as follows: [1]H NMR (400 MHz, CDCl$_3$, Bruker, Billerica, MA, USA,), δ (ppm): 7.68 (s, 1H), 7.00 (s, 1H), 5.51 (m, 2H), 5.16 (br, 1H), 4.02-3.90 (m, 6H), 3.85-3.76 (m, 6H), 3.25 (t, 2H), 1.67 (m, 2H), 1.23 (m, 9H), 0.67 (t, 2H). The corresponding results were also shown in Figure 1.

### 3.2. Photo-Cleavage of NVOC-U-PTEOS

Figure 2 shows the spectral evolution of NVOC-U-PTEOS methanol solution with the irradiation time. The peak around 350 nm was obviously decreased and red-shifted with photo-irradiation time until 2 h irradiation, indicating that the photo-cleavage reaction of NVOC-U-PTEOS occurred readily in methanol. This result is consistent with the previous observations of the spectral changes during photo-irradiation of accompanying NVOC-modified compounds [19,23]. According to the literature, the photo-cleavage reaction could occur as shown in Figure 2. Irradiation of the NVOC-U-PTEOS with UV light at 365 nm leads to a reduction of the specific NVOC absorbances at 300 and 350 nm, and at the same time to an obvious rise around 375 nm due to the formation of the protonated nitro product. And the decrease in the peak at 375 nm for significantly longer irradiation times can be explained by the formation of nitroso product. [35]. The UV spectra confirmed that the photo-cleavage of the prepared NVOC-U-PTEOS successfully occurred.

### 3.3. Morphologies of the Prepared Particles

Figure 3 shows the SEM and TEM images of the relative particles. The SEM images shown in Figure 3a,b indicate that both NVOC-U-PTEOS-SiO$_2$ microspheres and H$_2$N-SiO$_2$-NH$_2$ microspheres were monodispersed with the size of 450 nm, and the morphologies of NVOC-U-PTEOS-SiO$_2$ microspheres were similar to those of H$_2$N-SiO$_2$-NH$_2$, indicating that the irradiation of NVOC-U-PTEOS-SiO$_2$ microspheres did not affect their morphology. Although the diversity of morphology between two kinds of the microspheres was not obvious, their properties are significantly different. The difference in property between two kinds of microspheres was verified by contact angle. Obviously, NVOC-U-PTEOS-SiO$_2$ microspheres were hydrophobic due to their contact angle of 121°, while H$_2$N-SiO$_2$-NH$_2$ microspheres were hydrophilic because they were completely immersed in water. These results implied that the photo-cleavage reaction of NVOC-U-PTEOS could occur on the surface of NVOC-U-PTEOS-SiO$_2$ microspheres. To verify the asymmetry of NVOC-U-PTEOS-

SiO$_2$-NH$_2$ microspheres in composition, the interaction of the colloidal gold nanoparticles with NVOC-U-PTEOS-SiO$_2$-NH$_2$ microspheres and H$_2$N-SiO$_2$-NH$_2$ microspheres were performed. The preparation of gold nanoparticles by citric acid reduction has a negative charge characteristic, and the prepared NVOC-U-PTEOS-SiO$_2$-NH$_2$ microspheres have an amine group pKa of about 10. NVOC-U-PTEOS-SiO$_2$-NH$_2$ microspheres are positively charged in a neutral water medium. Therefore, through electrostatic interaction, gold nanoparticles can be adsorbed on the surface of NVOC-U-PTEOS-SiO$_2$-NH$_2$ microspheres.

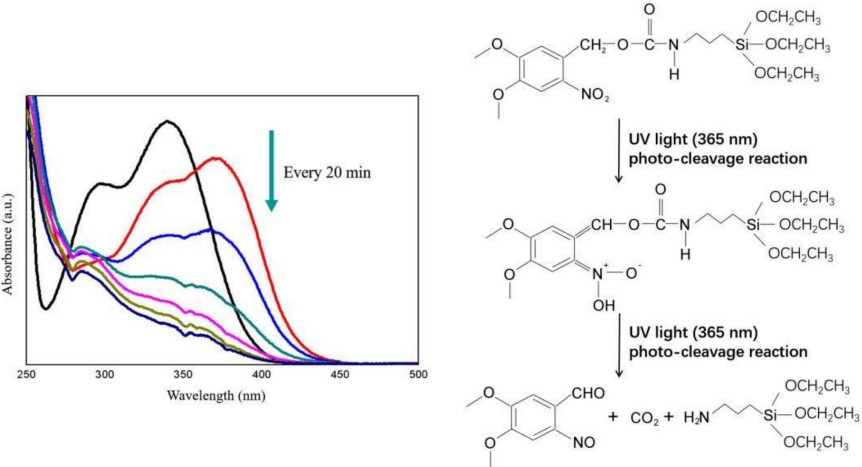

**Figure 2.** Change in UV spectra with time during irradiation of NVOC-U-PTEOS in methanol (**left**) and the corresponding reaction (**right**).

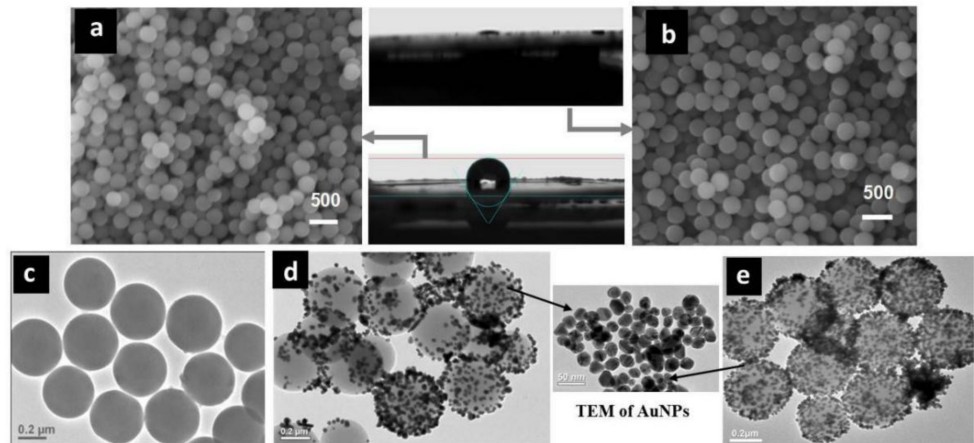

**Figure 3.** SEM images of (**a**) NVOC-U-PTEOS-SiO$_2$, (**b**) H$_2$N-SiO$_2$-NH$_2$; TEM images of (**c**) the formed monolayer of NVOC-U-PTEOS-SiO$_2$, (**d**) NVOC-U-PTEOS-SiO$_2$-NH$_2$ loaded gold nanoparticles, (**e**) H$_2$N-SiO$_2$-NH$_2$ loaded AuNPs.

The TEM images in Figure 3c indicated that the monolayer of NVOC-U-PTEOS-SiO$_2$ particles could form by using self-assembly. When one side of this monolayer was vertically irradiated, NVOC-U-PTEOS-SiO$_2$-NH$_2$ particles could form. As two sides of this monolayer were vertically irradiated, H$_2$N-SiO$_2$-NH$_2$ particles could form. As mentioned above, the citrate stabilized AuNPs with negatively charge could be absorbed on one side with the positively charged amino groups of NVOC-U-PTEOS-SiO$_2$-NH$_2$ microspheres [36]. Analogously, the AuNPs could be adsorbed on whole surfaces of H$_2$N-SiO$_2$-NH$_2$ particles. As expected, for the asymmetrically photo-irradiated microspheres, both AuNPs-covered and AuNPs-uncovered microspheres were found (Figure 3d), probably implying that the asymmetrically photo-irradiated microspheres possessed an asymmetrical distribution

of amino groups. In contrast, for the symmetrically photo-irradiated microspheres, all microspheres were homogeneously covered by AuNPs (Figure 3e), indicating that the symmetrically photo-irradiation of NVOC-U-PTEOS-SiO$_2$ microspheres have a symmetrical distribution of amino groups. In fact, the characteristics of distribution of amino groups on these kinds of microspheres result in their different properties. For the microsphere with asymmetrical distribution of amino groups, one side of the microsphere has hydrophilic amino groups, and the other side has hydrophobic 6-nitroveratryloxycarbonyl moieties, which makes the microsphere become amphiphilic. However, the microsphere with symmetrically distributed amino groups is hydrophilic. The characteristics mentioned above were confirmed in the following sections.

### 3.4. Behaviors of the Prepared Microspheres Dispersed in the Oil-Water Dual-Phase

Generally, the concordance in the hydrophilicity/hydrophobicity between a particle and a solvent is favorable to the suspension stability of the particle. As aforementioned, the asymmetrically photo-irradiated microspheres, NVOC-U- PTEOS-SiO$_2$-NH$_2$, are amphiphilic due to the microsphere with hydrophilic NH$_2$ groups in one side and hydrophobic 4, 5-dimethoxy-2-nitrobenzyl moieties. The symmetrically photo-irradiated microspheres, H$_2$N-SiO$_2$-NH$_2$, are hydrophilic, and NVOC-U-PTEOS-SiO$_2$ microspheres are hydrophobic. These differences can be proved by dispersed behaviors in the oil–water dual-phase system. Figure 4 shows photographs of H$_2$N-SiO$_2$-NH$_2$ microspheres (a), NVOC-U-PTEOS-SiO$_2$-NH$_2$ microspheres (b) and NVOC-U-PTEOS-SiO$_2$ microspheres (c) dispersed in a dual-phase of water and n-hexane system.

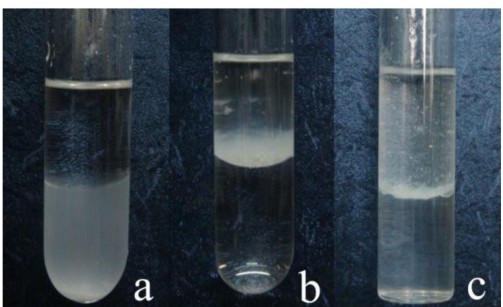

**Figure 4.** Photographs of different microspheres dispersed in dual-phase of water (lower) and n-hexane (upper) system. (**a**) H$_2$N-SiO$_2$-NH$_2$ microspheres, (**b**) NVOC-U-PTEOS-SiO$_2$-NH$_2$ microspheres, and (**c**) NVOC-U-PTEOS-SiO$_2$ microspheres.

As predicted, H$_2$N-SiO$_2$-NH$_2$ microspheres were dispersed only in lower phase water due to their hydrophilicity (Figure 4a). For NVOC-U-PTEOS-SiO$_2$ microspheres, they were dispersed only in upper phase n-hexane due to their hydrophobicity (Figure 4c). Interestingly, NVOC-U-PTEOS-SiO$_2$-NH$_2$ microspheres were located at the phase boundary between water and n-hexane (Figure 4b). Commonly, the surface of AJP is compartmentalized and has two parts exhibiting different wettability. Therefore, the Janus particle is similar to a classical surfactant in an interface property [37]. As a result, NVOC-U-PTEOS-SiO$_2$-NH$_2$ microspheres with amphiphilicity are favorable to disperse in the boundary between the hydrophilic solvent and the hydrophobic solvent. Based on the results shown in Figure 4, it could be concluded that the asymmetrically photo-irradiated microspheres have asymmetrical distribution of amino groups.

### 4. Conclusions

The SiO$_2$ particles with one side of hydrophilic amino groups and other side of hydrophobic 4, 5-dimethoxy-2-nitrobenzyl moieties could be obtained by using the vertically photo-irradiated SiO$_2$ particles surface immobilized the 6-nitroveratryloxycarbonyl group modified alkoxysilane via urethane functional group. The prepared particles process the feature of amphiphilic Janus particles (AJP). To our best knowledge, it is the first time that

AJP are prepared by utilizing the concept of the caged compounds. The major feature of this route was highly flexible because the chemical and physical properties of the prepared AJP could be easily tailored through reaction of amino groups with isocyanate, sulfonyl chloride, aldehyde, carbodiimide, acyl azide, anhydride, N-hydroxysuccinimide esters, imidoester, epoxide, and so forth. Based on the modification, AJP linked with various functional groups or chains could be obtained. Owing to the wide variety of applications of $SiO_2$ microspheres and the easy modification of primary amines, the novel $SiO_2$-based Janus microsphere can be extended to generate various functional Janus microspheres used in the desired applications. Therefore, the prepared AJP are a fascinating platform for the succeeding functionalization with other molecules and provide beneficial clues for the field of materials science.

**Author Contributions:** Conceptualization, W.L. and D.H.; methodology, D.H.; software, J.M.; validation, W.L., S.S. and D.H.; formal analysis, J.M.; investigation, W.L.; resources, D.H.; data curation, W.L.; writing—original draft preparation, W.L.; writing—review and editing, W.L. and D.H.; supervision, D.H.; project administration, D.H.; funding acquisition, D.H. and W.L. All authors have read and agreed to the published version of the manuscript.

**Funding:** This research was funded by the National Science Foundation of China (No.21173141 and 21303135); and funded by the China Postdoctoral Science Foundation (No.2019M653863XB).

**Data Availability Statement:** Not applicable.

**Conflicts of Interest:** The authors declare no conflict of interest.

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
