# Peer review of "Amphiphilic Janus Microspheres Prepared by Caged Photoactivatable Alkoxysilane"

_coatings, doi:10.3390/coatings12020198_

Round 1

Reviewer 1 Report

The present manuscript entitled “Amphiphilic Janus Microspheres Prepared by Caged Photoactivatable Alkoxysilane” Wei Li et al., describe the preparation of Amphiphilic Janus Particles (AJP) based on SiO2 microspheres and the surface of SiO2 microspheres were modified by photoactive alkoxysilane, which was synthesized by dealcoholization condensation of 6-nitroveratroyloxycarbonyl and isocyanatopropyl-triethoxysilane. Furthermore, the surface characteristics of Janus microspheres were confirmed by selective adsorption of golden nanoparticles as probes. It is a well-organized article and lack of major errors. The article is well-written; however, the quality of some of the figures can be improved in general. I recommend acceptance in the Coatings.

I advise the authors to take the following minor points into account while revising their manuscript.

Comment 1:  There are some typographical errors in the manuscript text, so authors need to correct them in the revised manuscript.

Comment 2: The abstract is poorly written, should be edited. It must summarize well the obtained results.

Comment 3: In the introduction section add some recent literature to strengthen the section.

Comment 4: The FT-IR and SEM analysis should be discussed with some more references in order to prepare a better discussions.

Comment 5: In the whole manuscript the authors must be taken care of the superscripts and subscripts and abbreviations.

Reviewer 2 Report

In this manuscript, Li et al. described a new method to prepare asymmetric particles based on the photo-cleavage of the nitroveratroyloxycarbonyl moiety. The latter was anchored to spherical SiO2 particles through a previous treatment with a home-made triethoxysilane derivative and subsequently engaged in a selective UV irradiation. The partial illumination of the particles monolayer allowed the functionalization of the upper hemisphere, keeping the protecting groups of hidden face. Eventually, the Janus particles were characterized by SEM and compared with its symmetrical counterparts non- and fully-deprotected.

In summary the research is promising and well conducted, thus I would recommend it for publication in coatings after addressing some queries:

  • I am not totally sure that the concept Janus polymer is really common or used within the community. Pierre-Gilles de Gennes talked about ‘Janus grains’ and concepts like Janus particles, colloids, dendrimers, etc., are really well known, but please consider if Janus polymer is correct or not.
  • Owing to the relevance of photo-irradiation in the manuscript, authors should cite recent works using photo-polymerization of complex emulsions to fabricate Janus particles, e.g., 10.1063/1.5018207, 10.1021/acsami.1c07259.
  • Please rephrase or explain the following sentence: ‘photo-cleavable protecting groups… are very suitably used in the preparation of AJP’. In the previous sentences it is said that they have been ‘scarcely reported’, would it be more correct that they could be suitably used? On the other hand, if they are suitably used, add some references.
  • In Scheme 2, is it really contributing scientifically the Janus pic? I would understand it in a TOC, but from my point of view it could be removed from this scheme.
  • Regarding the purification of TEOS, please indicate the method as experiments can be reproduced by the reader.
  • A mistake in Scheme 4 should be corrected: the SiO2 particle should be bounded to a Si atom not an oxygen.
  • The sentence ‘This peak disappeared and simultaneously, a new peak assigned for the same a-methyl group in the structure of the nitroso ketone appeared,’ is doubtful and hard to understand, please rephrase. In any case, absorption peaks are not assigned to functional groups, but to molecular transitions. Introduce the wavelength of the peak while taking.
  • Could the authors explain the bathochromic shift of the Abs maximum band after the photo-cleavage and in general the behaviour of the UV-vis spectra? It seems straightforward that the band disappears owing to the consumption of the starting material but the red shift could be ascribed to the formation of an intermediate. The absence of an isosbestic point is in agreement with the formation of a reaction intermediate.
  • Could the authors explain why in figure 3 d) and e) are magnifications of the same micrograph? Should not the different materials come from different batches?
  • Regarding the surfactant character of the Janus particles, I would recommend the authors to test their NVOC-U-PTEOS-SiO2-NH2 to prepare O/W emulsions, proving their real potential as surfactant materials. Besides the surfactant strength can be analysed via simple experiments such pendant drop technique.
  • In Conclusions, I would be careful with the claim: ‘this route was highly flexible…’, since none of the later mentioned functionalization were proved for the particles prepared in the manuscript.

Reviewer 3 Report

coatings-1557276-peer-review-v1

 Amphiphilic Janus Microspheres Prepared by Caged Photoactivatable Alkoxysilane

Wei Li 1,2,*, Daodao Hu 2, Jinxin Ma 3 and Shukun Shen

Dear Editor,

I looked at the manuscript. The Authors present a method fabricating functional sub-micron sized SiO2 amphiphilic janus particles with chemically reactive amino group. The photoactive alkoxysilane NVOC-U-PTEOS is utilized as surface modification reagent on hydrophilic SiO2 particles through the dealcoholization condensation between hydroxyl groups and ethoxyl groups. Then the SiO2 monodisperse layer is vertically photo-irradiated for a given time. Finally, SiO2 AJP with amphiphilicity could be obtained from one side with hydrophobic 6-nitro veratroyl oxy carbonyl groups and other side with hydrophilic amino groups. This work is interesting. The manuscript was well documented and well written. It can be acceptable after minor revision.

Comments

Scheme 4. Reaction between NVOC-U-PTEOS and SiO2. Something is wrong with the chemical formula of the product! Is that Si, instead of “O”?

It is not understandable what happens when gold nanoparticles interaction of the NVOC-U-PTEOS-SiO2-NH2 microspheres; this needs reaction scheme.

In Figure 4, dual phase formation should be explained clearly.

This work is about amphiphilic polymers. The authors should mention about the amphiphilic polymers in introduction section. It is advised for the authors to review the following articles:

DOI 10.1007/s00289-009-0211-3

Adv. Mater. 1998, 10, No. 3, 195-217

In the table content figure, two faces of a man head is not understandable in this work.
